# Efficient Removal of Cd(II) Using SiO_2_-Mg(OH)_2_ Nanocomposites Derived from Sepiolite

**DOI:** 10.3390/ijerph17072223

**Published:** 2020-03-26

**Authors:** Zhendong He, Bozhi Ren, Andrew Hursthouse, Zhenghua Wang

**Affiliations:** 1Hunan Provincial Key Laboratory of Shale Gas Resource Exploitation, Xiangtan 411201, China; ZhendongHe@126.com (Z.H.); wzh@hnust.edu.cn (Z.W.); 2School of Civil Engineering, Hunan University of Science and Technology, Xiangtan 411201, China; 3School of Computing, Engineering & Physical Sciences, University of the West of Scotland, Paisley PA1 2BE, UK

**Keywords:** cadmium, composite modification, nanomaterial, sepiolite, adsorption

## Abstract

The pollution of Cadmium (Cd) species in natural water has attracted more and more attention due to its high cumulative toxicity. In the search for improved removal of cadmium from contaminated water, we characterized uptake on a recently identified nanomaterial (SiO_2_-Mg(OH)_2_) obtained by subjecting sepiolite to acid-base modification. The structural characteristics of SiO_2_-Mg(OH)_2_ were analyzed by means of SEM-EDS, Fourier Transform Infra-Red Spectroscopy (FTIR) and Powder X-ray Diffraction (PXRD). Static adsorption experiments were carried out to evaluate the effect of contact time, temperature, amount of adsorbent, and pH-value on the adsorption of Cd(II) by SiO_2_-Mg(OH)_2_. The results show that the pore structure of SiO_2_-Mg(OH)_2_ is well developed, with specific surface area, pore size and pore volume increased by 60.09%, 16.76%, and 43.59%, respectively, compared to natural sepiolite. After modification, the sepiolite substrate adsorbs Cd(II) following pseudo-second-order kinetics and a Langmuir surface adsorption model, suggesting both chemical and physical adsorption. At 298 K, the maximum saturated adsorption capacity fitted by Sips model of SiO_2_-Mg(OH)_2_ regarding Cd(II) is 121.23 mg/g. The results show that SiO_2_-Mg(OH)_2_ nanocomposite has efficient adsorption performance, which is expected to be a remediation agent for heavy metal cadmium polluted wastewater.

## 1. Introduction

In recent years, due to the high toxicity, non-degradability, enrichment of heavy metals and the potential threat to human health through water pollution, the problem of heavy metal pollution has attracted more and more attention [1,2]. The heavy metal cadmium(Cd) is a non-essential element of the human body and is widely used in electroplating, batteries, smelting and chemical production processes, resulting in a considerable amount of cadmium in wastewater, waste gas and waste residue [3]. After entering the environment, it cannot be biodegraded. It is enriched and transferred in the body through the food chain. When the concentration of cadmium reaches a certain level, cadmium poisoning will occur, and it can even cause various diseases of the human body [4,5]. Most of the cadmium in water exists in the form of metal cation Cd(II).At present, common methods for removing Cd(II) are ion exchange [6], chemical precipitation [7], membrane separation [8], and adsorption [9]. Adsorption method is widely used because of its advantages such as simple operation, simple process, and abundant sources of adsorbent. Natural clay minerals have good internal pore structure, large surface area, strong chemical adsorption, and abundant reserves, so they are expected to become cheap adsorbents for water treatment [10].

Sepiolite (Sep) is a natural fibrous magnesium-rich silicate clay mineral. Its crystal form is orthorhombic and its standard crystal chemical formula is: Mg_8_Si_12_O_30_(OH)_4_(OH_2_)_4_·8H_2_O. The overall structure of sepiolite is the crystal structure of 2:1 type clay minerals composed of three pyroxene single chains, which can be specifically divided into upper, middle, and lower three layers. The upper and lower layers are continuous silicon-oxygen tetrahedrons, while the middle layer is a discontinuous magnesium-oxygen octahedron. It contains “zeolitic water” and some exchangeable Mg^2+^ and Ca^2+^ ions in the nanostructured channels which are measured to be about 1.06 × 0.37 nm^2^ in cross section [11]. This structure determines the adsorption performance and ion exchange capacity, so it is widely used in the study of heavy metal adsorption [12,13]. The theoretically estimated specific surface area of sepiolite can reach 900 m^2^/g, which are easy to obtain at relatively low price and has potential in the treatment of polluted wastewater, but due to various factors, the actual measured specific surface area value is relatively small. Therefore, appropriate modification of sepiolite is needed [14,15]. After sepiolite treatment with acid, the surface area increased about twice, and opened up the pore structure increasing the number of active sites [16]. The effectiveness of the three acids followed the order HCl > HNO_3_ > H_2_SO_4_ [17]. The acidification step sequentially removes Mg from the mineral structure and ultimately it can be completely converted into an ionic state and an amorphous silica gel can be generated [18]. However this extreme treatment is not appropriate for removal of pollutants as the presence of exchangeable Mg is a requirement for pollutant removal and excessive acidification reduces adsorption efficiency for Cd(II) in wastewater [19].

In order to overcome this disadvantage and increase the specific surface area, synthetic composites are considered to be one of the most effective methods for the modification of sepiolite. The main component of brucite is Mg(OH)_2_, which has proven to be a good adsorbent for Cd(II) [20,21]. A common method to suppress the agglomeration of nanomaterials is by loading the nanomaterials on the mesoporous materials [22]. Yuan et al. [23,24] loaded nano-magnetite on montmorillonite, and compared with unloaded materials. The effect on removal of hexavalent chromium was significantly improved. Chen et al. [25] studied and summarized the latest preparation strategies, properties and applications of magnetic nanoparticle/clay mineral (MNP/CM) nanocomposites and proposed that MNP/CM nanocomposites have excellent magnetic properties, stability, adsorption, catalytic properties, biocompatibility and good application prospect. Yao et al. [26] carried out sequential acid-base composite modification of sepiolite to prepare SiO_2_-Mg(OH)_2_ nanocomposites, which exhibited high removal efficiency toward Gd(III), Pb(II) and Cd(II). Sepiolite is a natural nanomaterial with a relatively stable structure and an excellent carrier for other nanoadsorbents [27,28]. Therefore, it can effectively inhibit the agglomeration of nanostructured Mg(OH)_2_.

In this study, nanocomposites (SiO_2_-Mg(OH)_2_) were obtained by sequential acid-base treatment of sepiolite. The physicochemical characterization of SiO_2_-Mg(OH)_2_ was conducted by SEM-EDS, FTIR, PXRD and other conventional methods. Batch experiments were carried out to study influence of pH, sorbent dosage, adsorption isotherm, kinetic behavior and adsorption thermodynamics to provide a theoretical basis for the treatment of cadmium pollution in water.

## 2. Materials and Methods

### 2.1. Materials

Sepiolite was purchased from Sigma Reagent Company and was used without further purification. The sepiolite was pulverized by mortar grinding, sieved through a 300-mesh sieve, dried and put into a vacuum dryer before use. Cadmium powder, nitric acid, hydrochloric acid, and ammonia were purchased from Sinopharm Chemical Reagent Co., Ltd. (Shanghai, China). All reagents were of analytical grade and were used without further purification.

### 2.2. Synthesis of the SiO_2_-Mg(OH)_2_ and Characterization Methods

The preparation process of the material refers to the research of Yao et al. [26]. Specific steps are as follows: 4.0 g of sepiolite was first added to 100 mL of 20% (*v*/*v*) HCl solution, performing acid activation for 24 h with a stirring speed of 300 r/min, and then ammonia water was added dropwise to the solution under magnetic stirring until the pH of the solution reached 10.0. After stirring for 2 h, the suspension was centrifuged at 10,000 rpm for 10 min, and then washed several times with deionized water and absolute ethanol. The precipitate was finally collected and dried under vacuum at 60 °C.

Prior to adsorption of Cd(II) in simulated wastewater, sepiolite and SiO_2_-Mg(OH)_2_ derived from sepiolite were characterized. The surface morphology and structural characteristics of the samples were analysed using a scanning electron microscope with energy dispersive spectroscopy (SEM-EDS, JSM-6700F, JEOL, Akishima, Japan). Powder X-ray diffraction (PXRD) was obtained using an X-ray diffractometer (D/Max 2500, Rigaku, Japan). Specific surface area and pore diameter were conducted with a specific surface and porosity analyzer (ASAP 2020M, Micromeritics Instrument Ltd., Shanghai, China). Infrared spectrum was analyzed by Fourier transform infrared spectrometer (Nicolet 380, Thermo Electron Instruments Co., Ltd., Shanghai, China).

### 2.3. Batch Adsorption Experiments

#### 2.3.1. Adsorption Equilibrium and Kinetic Experiments

Dissolved the cadmium powder with concentrated nitric acid to make a mother liquor of 1 g/L, pH 3. In the experiment, the simulated wastewater containing different concentrations of cadmium was diluted by the mother liquor multiples. Sepiolite was added to the simulated cadmium wastewater and oscillating perform adsorption at a rate of 150 r/min according to different experimental requirements and conditions. After passing through a 0.45 μm filter, the concentration of Cd(II) in the filtrate was measured using an atomic absorption spectrophotometer (Ultra TAS-990, Purkinje General Instrument Co., Ltd., Beijing, China). Cd(II) adsorption efficiency *η* (%) and equilibrium adsorption capacity (*q_e_*) are calculated by the following formulas:(1)$$\eta =\frac{C_{0}-C_{e}}{C_{0}}$$
(2)$$q_{e}=\left(\frac{C_{0}-C_{e}}{m}\right)\cdot V$$
where *C*_0_ and *C*_e_ are the initial and equilibrium concentrations (mg/L) of Cd(II) respectively, *m* is the mass of the adsorbent (g), and V is the volume of the solution (L).

For the kinetic studies, 50 mL Cd(II) solutions (50, 100 and 150 mg/L) were added to a 100 mL capped Erlenmeyer flask with HCl (0.1–1 mg/L) and NaOH (0.1–1 mg/L) to adjust the pH value to 7.0, put it in a constant temperature air bath shaker, and add 0.5 g/L of SiO_2_-Mg(OH)_2_ when the temperature reaches 45 °C. After shaking for a certain time (5–720 min) at constant temperature the adsorption of SiO_2_-Mg(OH)_2_ for heavy metal cation Cd(II) was recorded. The adsorbed amount is plotted as a function of the adsorption time.

#### 2.3.2. Effect of Adsorbent Dosage and pH-Value

Under the experimental conditions of a temperature of 35 °C, a solution volume of 50 mL, an initial concentration of Cd(II) of 100 mg/L, an initial pH value of 6 and a oscillation adsorption time of 240 min, the SiO_2_-Mg(OH)_2_ was investigated. When the adsorbent dosage (0.2–2.5 g/L) is different, it will affect the adsorption of heavy metal Cd (II) respectively.

Varying the pH value between 2 and 7 by keeping the other parameters constant, the effect of the pH value on Cd(II) adsorption is evaluated.

#### 2.3.3. Adsorption Isotherms

Adsorption isotherm experiments at three different temperatures (25 °C, 35 °C and 45 °C) were carried out by changing the initial concentration of Cd(II) from 30 to 200 mg/L. The procedure was like the procedure above except for vibrating for 4 h. Isotherm models such as the Freundlich, the Langmuir and the Sips isotherm were employed to interpret the adsorption isotherm data. 

## 3. Results and Discussion

### 3.1. Characterization of SiO_2_-Mg(OH)_2_ Nanocomposites

#### 3.1.1. SEM-EDS Analysis

Figure 1a,b shows the SEM-EDS images of the sepiolite and SiO_2_-Mg(OH)_2_. It can be seen from Figure 1a that the structure of the sepiolite material is a thin and straight rod-like fiber structure. In Figure 1b, a scanning electron microscope image of the treated sepiolite—SiO_2_-Mg(OH)_2_ material after being treated with hydrochloric acid and ammonia is shown. It can be seen that compared to the original the surface of the composite material is uneven, and shows broken nanofibers in a messy block like structure, showing agglomeration and increased pore volume, indicating that the sepiolite-like magnesium octahedron structure is destroyed, the silicon-oxygen tetrahedron is retained, and SiO_2_ colloids are shown formed [29]. The EDS in Figure 1 confirms the main constituent elements of the sepiolite as well as of the SiO_2_-Mg(OH)_2_ material, which are O, Si, Mg. The signal for magnesium in SiO_2_-Mg(OH)_2_ is stronger than for pure sepiolite due to the relative enrichment during alkali treatment to form a magnesium-containing compound on the surface.

#### 3.1.2. N_2_ Adsorption-Desorption Isotherm and Pore Size Distribution

The N_2_ adsorption-desorption isotherm and pore size distribution for sepiolite and the SiO_2_-Mg(OH)_2_ composite is shown in Figure 1c,d. Both materials show type IV adsorption [30] and their pore size distribution is concentrated in 2–50 nm diameter, which indicates that there is a large number of mesopores in the sample, which is conducive to adsorption. The specific surface area of the SiO_2_-Mg(OH)_2_ composite material (458.29 m^2^/g) is significantly higher than that of sepiolite (301.66 m^2^/g), and the average pore diameters of the SiO_2_-Mg(OH)_2_ composite material and sepiolite calculated by BJH are 3.97 nm and 3.40 nm, respectively, the pore volumes are 0.56 and 0.39 cm^3^/g. Compared with the original sepiolite, they have increased by 60.09%, 16.76%, and 43.59%. After modification, surface properties have been greatly improved due to removal of Mg^2+^ from the structure by continuous alkali treatment, from which Mg(OH)_2_, becomes attached to the acidified sepiolite. Because of this, the specific surface area and pore volume of sepiolite are further increased.

#### 3.1.3. FTIR Analysis

In Figure 1e the infrared spectrum of sepiolite and SiO_2_-Mg(OH)_2_ composite material are overlain. The wave numbers range between 1300–900 cm^−1^, it is a wide and strong stretching vibration band produced by Si-O-Si [31,32]. The peak at 3569 cm^−1^ corresponds to the stretching of OH connected to Mg^2+^. The absorption peak at 485 cm^−1^ originates from the vibrational absorption of the Si-O-Mg bond in the O-Mg octahedron within the sepiolite crystal. These peaks are absent from the SiO_2_-Mg(OH)_2_ spectrum, demonstrating the breakdown of the sepiolite framework. The infrared spectrum for SiO_2_-Mg(OH)_2_ is dominated by the characteristic peaks for SiO_2_ and Mg(OH)_2_. The vibration absorption peaks for Si-O-Si are at 1108 cm^−1^ and 790 cm^−1^. The absorption peaks around 3429–3500 cm^−1^ are the stretching vibrations of intermolecular hydrogen bonds O-H; the absorption peaks at 689 cm^−1^ and 641 cm^−1^ correspond to the absorption vibrations of Mg-OH bonds. The vibration band in the range of 700–600 cm^−1^ is different from that of SiO_2_, which may be related to the brucite subsequently formed.

#### 3.1.4. PXRD Analysis

X-ray diffraction was used to test the internal crystal structure of sepiolite and SiO_2_-Mg(OH)_2_ composites. During the test, the 2θ angle range was 5° to 60°. It can be seen from the Sep spectrum in Figure 1 that significant diffraction peaks appear at 7.3°, 11.8°, 19.7°, 20.4°, 23.7°, 26.4°, and 35.0°. According to the standard card number ((JCPDS: 13-0595) for (110), (130), (060), (131), (260), (080), and (371)) they correspondto sepiolite. The PXRD pattern for the SiO_2_-Mg(OH)_2_ composite is less distinct with a humped baseline and few characteristic diffraction peaks at (2θ) 20.9°, 26.6°, and 50.1°, which are consistent with the presence of SiO_2_ (JCPDS: 46-1045). In addition smaller peaks 2θ are found at 18.5° and 50.8°, which are consistent with Mg (OH)_2_ (JCPDS: 44-1482). It is consistent with the results of the infrared spectroscopy, showing more amorphous phase produced during treatment. The schematic diagram of material generation is shown in Figure 2.

### 3.2. Adsorption Kinetics

Non-linear fitting was performed using pseudo-first-order (3) and pseudo-second-order (4) equations. The formulae are used as follows:
(3)$$q_{t}=q_{e}\left(1-e^{-K_{1}t}\right)$$
(4)$$q_{t}=\frac{K_{2}q_{e}^{2}t}{1+K_{2}q_{e}t}$$
where *t* is the adsorption time (min); *K*_1_ is the pseudo-first-order kinetic equation rate constant (min^−1^); *K*_2_ is the pseudo-second-order kinetic equation rate constant (g·mg^−1^·min^−1^); *q_t_* and *q_e_* are the Cd(II) adsorption amount (mg·g^−1^) at time *t* and adsorption equilibrium, respectively.

As shown in Figure 3, the adsorption capacity of SiO_2_-Mg(OH)_2_ for Cd(II) first increased rapidly and then stabilized: within 1 h, the adsorption rate of the material is very large; within 1–3 h, the adsorption rate of the material gradually decreased and reached equilibrium in about 4 h. The maximum adsorption of Cd(II) at an initial concentration of 150 mg·g^−1^ is 92.76 mg·g^−1^, which is much larger than the original sepiolite adsorption capacity for Cd(II) [33]. This is because the SiO_2_-Mg(OH)_2_ composite has a larger specific surface area, so the number of surface active adsorption sites after the modification of the original sepiolite increases, and the adsorption capacity is enhanced. The initial rapid adsorption might be due to the external diffusion of Cd(II) transfer into the porous structure fast enough. This process is mainly ion exchange and physical adsorption. However, with the increase in time, the free active sites on the surface of the adsorbent for Cd(II) adsorption decreased, the concentration gradient of Cd (II) in the solution and the surface of the adsorbent decreased, and the driving force for adsorption decreased, so for Cd(II) the resistance of adsorption by diffusion into the adsorbent increases, and the adsorption rate gradually decreases [34].

From Table 1, it can be seen that the *R*^2^ value of SiO_2_-Mg(OH)_2_ for Cd (II) adsorption by quasi-second-order kinetics is above 0.99, which is greater than the *R*^2^ value of quasi-first-order kinetics. On this basis it can be inferred that the adsorption of Cd (II) by SiO_2_-Mg(OH)_2_ is dominated by chemisorptions, and true chemical bond formation between the adsorbent and Cd (II) [35].

### 3.3. Effect of SiO_2_-Mg (OH)_2_ Dosage and pH on Cd (II) Removal

The study of the effect of the adsorbent dose and pH on the adsorption performance and the maximum adsorption effect of the adsorbent on different metal ions has obvious practical significance for the industrial application of the adsorbent. It can be seen from Figure 4 that as the amount of SiO_2_-Mg(OH)_2_ added increases, the removal rate of Cd (II) also increases, but the rate of increase slows and finally stabilizes. When the dose is 2 g/L, the removal rate reaches 90.88%. At the same time, with the increase of the adsorbent, the adsorption capacity showed a trend of first increase and then decrease. When the amount of adsorbent is 0.4 g/L, the maximum adsorption amount reaches 71.88 mg/g. This is because with the increase of the amount of adsorbent added, the number of adsorption sites is increasing, so that the removal rate increases rapidly at the beginning. However, when the amount of the adsorbent exceeds a certain concentration, some of the adsorption sites of the adsorbent cannot be completely occupied. As a result, the amount of Cd (II) adsorbed by the adsorption unit site is lower, which is shown by a decrease in the adsorption unit.

In the pH range from 2 to 7 (Figure 4), the amount adsorbed and removal of Cd (II) increased with the increase of pH. The adsorption effect is poorest at pH 2, because the lower the pH is, the higher the H^+^ content, the more intense the competition between H^+^ and Cd^2+^ for SiO_2_-Mg(OH)_2_ surface adsorption sites. The optimal pH was 7.0, and the maximum adsorption amount and removal rate were 69.48 mg/g and 34.74%, respectively. When pH > 7, OH^-^ in the solution will bind to Cd^2+^ and exist as Cd(OH)^+^ or Cd(OH)_2_, which will cause precipitation and inhibit adsorption [36].

### 3.4. Adsorption Isotherm

Three adsorption models (Figure 5) were used to study uptake data: Langmuir (5), Freundlich (6), and Sips (L-F) (7) were used for nonlinear fitting of adsorption isotherms at different temperatures (25 °C, 35 °C, and 45 °C).
(5)$$q_{e}=\frac{K_{L}Q_{m}C_{e}}{1+K_{L}C_{e}}$$
(6)$$q_{e}=K_{F}C_{e}{}^{1/n}$$
(7)$$q_{e}=\frac{Q_{m}K_{s}C_{e}^{\beta }}{1+K_{S}C_{e}^{\beta }}$$
where q_e_ is the equilibrium adsorption amount (mg/g); *C_e_* is the equilibrium concentration (mg/L); *Q_m_* is the saturation adsorption amount (mg/g); K_L_ (L·mg^−1^), K_F_ [mg·g^−1^· (L·mg^−1^)^1/n^] and K_S_ [(L·mg^−1^)^β^] are Langmuir adsorption constant, Freundlich adsorption constant, and Sips adsorption constant, respectively; 1/n is a constant for adsorption strength, which varies with the heterogeneity of the material; β is the Sips isotherm index [37].

It can be seen from Table 2 that the *R*^2^ values of Langmuir and Sips isotherms are higher than the *R*^2^ values of the Freundlich isotherms at different temperatures, and both can better describe the adsorption process of Cd (II). The Sips model has the best correlation. Langmuir isotherm is suitable for monolayer adsorption with uniformly distributed adsorption sites on the surface of the adsorbent and there is no interaction between adjacent sites and adsorbent particles, while the Freundlich adsorption isotherm model is an empirical formula for multi-layer adsorption [38,39]. Therefore, it can be speculated that the adsorption of Cd(II) at different temperatures is mainly a uniform monolayer adsorption process.

The Sips isothermal model is a combination of Langmuir and Freundlich equations, and is suitable for describing the adsorption of monolayer heterogeneous surfaces on heterogeneous adsorption systems under various pressures. At lower metal ion concentrations, it is closer to the Freundlich isotherm; while at high concentrations, it describes monolayer adsorption in a similar way to the Langmuir isotherm. The higher the Sips constant K_s_ value, the stronger the bond between the adsorbate and the active site of the adsorbent [33]. It can be seen that as the temperature increases, the adsorption capacity of SiO_2_-Mg(OH)_2_ becomes higher. An increase of the adsorption capacity with increasing temperature is typical for chemisorption processes.

According to the Sips model, the maximum adsorption capacity of Cd(II) by SiO_2_-Mg(OH)_2_ is 121.23 mg/g at 25 °C. According to Table 3, it can be seen that SiO_2_-Mg(OH)_2_ has compared with similar adsorbents very good adsorption performance.

### 3.5. Thermodynamic Parameters

Thermodynamic parameters (the standard free energy Δ*G*°, standard enthalpy Δ*H*° and standard entropy Δ*S*°) at three different temperatures were calculated by Equations (8) and (9), respectively.
(8)$$\Delta G{}^{\circ}=-RT\cdot {\mathrm{lnK}}_{0}$$
(9)$$\ln K_{0}=\frac{\Delta S{}^{\circ}}{R}-\frac{\Delta H{}^{\circ}}{RT}$$
where *K*_0_ is the distribution coefficient obtained from the Langmuir isotherm (L/mol).

With ln*K*_0_ plotted against *1/T*, the intercept and slope obtained from the linear regression analysis can be used to calculate *ΔS°* and *ΔH°*, respectively. And *ΔG°* can be calculated directly from Equation (8).

As shown in Table 4, when the temperature increases from 298 K to 318 K, the adsorption of Cd(II) on SiO_2_-Mg(OH)_2_ has Δ*G*° < 0 and Δ*H*° > 0, indicating that the adsorption is a spontaneous endothermic reaction. Δ*H*° is mainly physical adsorption between 0–80 kJ·mol^−1^, and chemical adsorption between 80–800 kJ·mol^−1^ [43]. In this study, the Δ*H*° of SiO_2_-Mg(OH)_2_ was 29.71 kJ·mol^−1^ between 298 and 318 K, therefore the adsorption process of Cd(II) belongs to physical adsorption. The adsorption mechanism may be electrostatic force and pore filling. Combining with the results of kinetic adsorption, it can be seen that there is physical and chemical adsorption of SiO_2_-Mg(OH)_2_ for Cd(II), but mainly chemisorption. lnK_0_ increases with increasing temperature, indicating that temperature rise is conducive to the progress of adsorption.

## 4. Conclusions

The SiO_2_-Mg(OH)_2_ nanocomposite based on sepiolite has developed pores and a large specific surface area and can be used as a good material for adsorbing Cd(II). The adsorption was dependent mainly on the initial solution pH, whereas the optimal Cd(II) adsorption was achieved at pH 7. Adsorption kinetics followed a pseudo-second-order kinetic model. The adsorption isotherm data of the nanocomposite adsorption were successfully described by the Sips isotherm model with a maximum adsorption capacity of 121.23–141.49 mg/g at 25–45 °C.

Therefore, this kind of nanocomposite material can effectively remove Cd(II) in water, and has great application potential in remediation of heavy metal cadium polluted wastewater.

## Figures and Tables

**Figure 1 ijerph-17-02223-f001:**
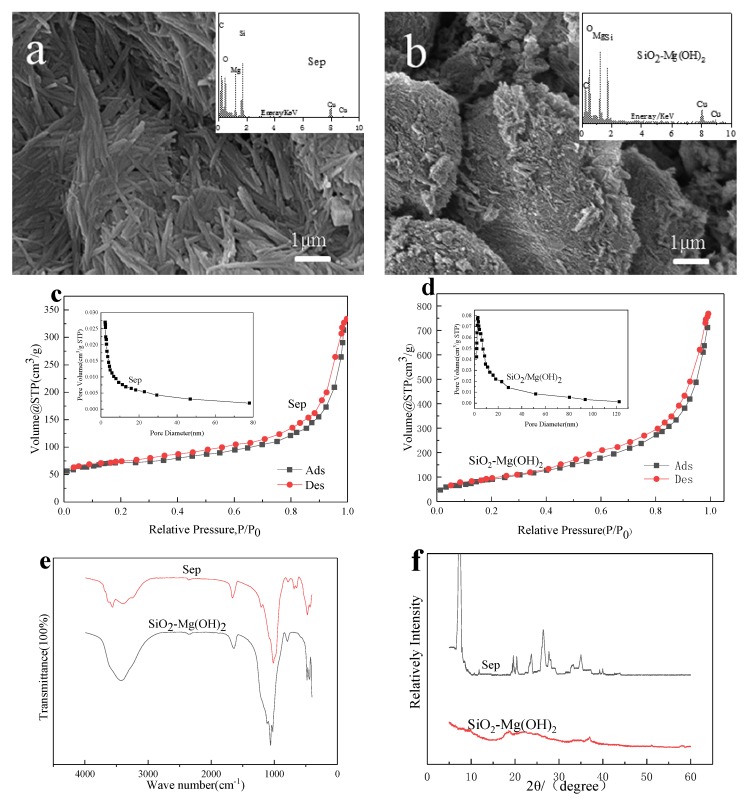
Characterization of Sep and SiO_2_-Mg(OH)_2_ composite. SEM-EDS image of sepiolite (**a**) and SiO_2_-Mg(OH)_2_ composite (**b**). BJH pore size distribution and N_2_ adsorption-desorption isotherm of sepiolite (**c**) and SiO_2_-Mg(OH)_2_ composite (**d**). FTIR spectrum (**e**) and PXRD (**f**) of sepiolite and SiO_2_-Mg(OH)_2_ composite.

**Figure 2 ijerph-17-02223-f002:**
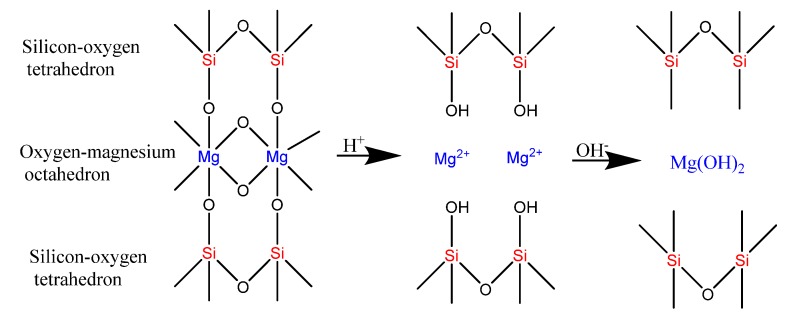
Schematic diagram of SiO_2_-Mg(OH)_2_ nanomaterial generation.

**Figure 3 ijerph-17-02223-f003:**
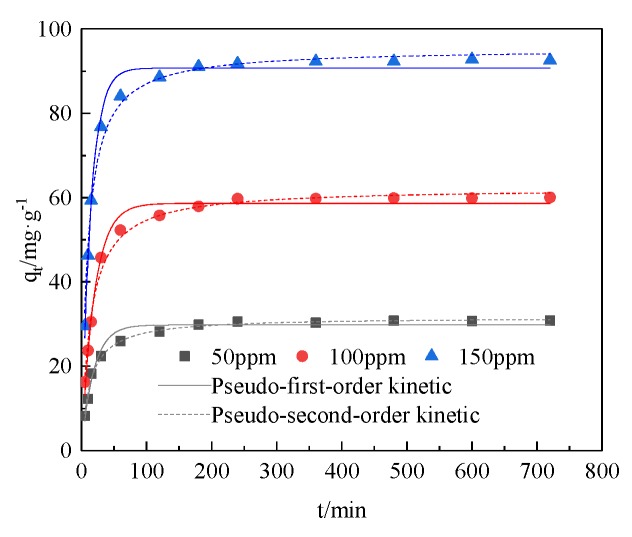
Adsorption kinetics of SiO_2_-Mg(OH)_2_.

**Figure 4 ijerph-17-02223-f004:**
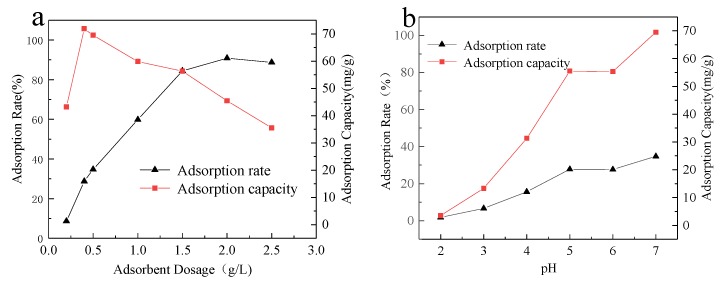
Effect of different dosage of adsorbent (**a**) and pH change (**b**) on Cd (II) adsorption.

**Figure 5 ijerph-17-02223-f005:**
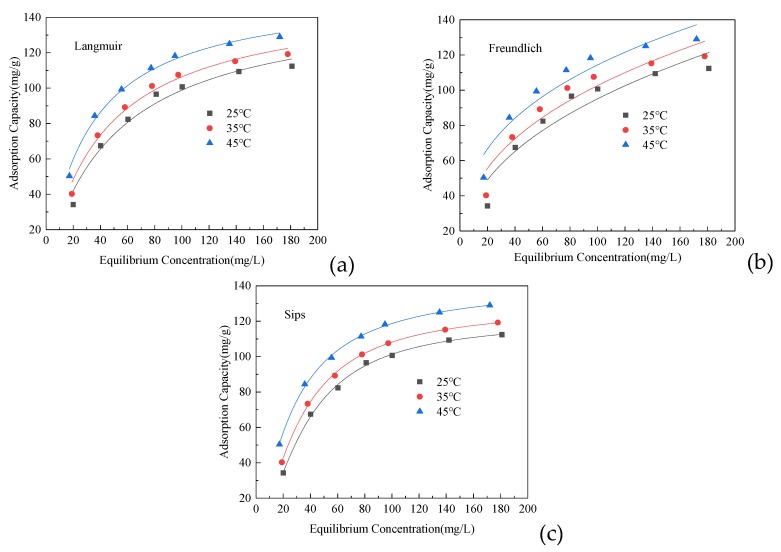
Equilibrium isotherms of Cd (II) adsorption onto SiO_2_-Mg (OH)_2_. (**a**) Langmuir; (**b**) Frundlich; (**c**) Sips.

**Table 1 ijerph-17-02223-t001:** Kinetic parameters of adsorption on Cd (II).

C_0_/mg·L^−1^	Pseudo-First-Order Kinetic Parameters			Pseudo-Second-Order Kinetic Parameters		
	q_e_/mg·g^−1^	K_1_/min^−1^	R^2^	q_e_/mg·g^−1^	K_2_/min^−1^	R^2^
50	29.8092	0.0537	0.9717	31.6627	0.0024	0.9930
100	58.6236	0.0505	0.9863	62.2908	0.0012	0.9910
50	90.7180	0.0699	0.9876	95.3443	0.0010	0.9909

**Table 2 ijerph-17-02223-t002:** Langmuir, Freundlich and Sips isotherm parameters for Cd (II) adsorption by SiO_2_-Mg (OH)_2._

T/K	Langmuir			Freundlich			Sips			
	K_L_/L·mg^−1^	Q_m_/mg·g^−1^	R^2^	K_F_/L·mg^−1^ (L·mg^−1^)^1/n^	n	R^2^	K_s_/(L·mg^1^)^β^	Q_m_/mg·g^−1^	β	R^2^
298	0.0195	149.5368	0.9760	14.2767	2.4287	0.9024	0.0035	121.2328	1.5806	0.9982
308	0.0234	151.9684	0.9840	17.7933	2.6284	0.9103	0.0063	128.9022	1.4609	0.9992
318	0.03115	155.7178	0.9913	24.3692	2.9791	0.9228	0.0152	141.4869	1.2674	0.9990

**Table 3 ijerph-17-02223-t003:** Comparison of Cd(II) adsorption by various adsorbents.

Sorbents	Qm(mg/g)	Experimental Conditions	Refenrences
TiO_2_/lignin	22.44	pH 5.0; T 20 °C	[36]
Succinic anhydride modified maize straw	196.1	pH 5.8; T 20 °C	[40]
S-ligand tethered cellulose nanofibers	92.17		[41]
Peanut shells	55.42		[42]
SiO_2_-Mg(OH)_2_	121.23	pH 7.0; T 25 °C	This work

**Table 4 ijerph-17-02223-t004:** Thermodynamic parameters for Cd(II) adsorption by SiO_2_-Mg(OH)_2._

Temperature (K)	lnK_0_	Δ*G*° (KJ·mol^−1^)	Δ*H*° (KJ·mol^−1^)	Δ*S*° (KJ·mol^−1^·K^−1^)
298	7.857	−19.47	29.71	0.1647
308	8.154	−20.88		
318	8.613	−22.77

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
