# Peer review of "Efficient Removal of Cd(II) Using SiO2-Mg(OH)2 Nanocomposites Derived from Sepiolite"

_ijerph, 2020, doi:10.3390/ijerph17072223_

Round 1

Reviewer 1 Report

In this paper, a new nano adsorbent (SiO2 -Mg(OH)2 )was studied, which was modified with acid and base. The morphology and structure of (SiO2 -Mg(OH)2 ) were characterized by SEM-EDS, FTIR and PXRD. (SiO2 -Mg(OH)2 ) has good adsorption property for Cd2+, but the following problems are suggested to be corrected:

1.The mechanism of ion exchange between adsorbent and Cd2+ need to be described in detail.

  1. How to consider the disadvantage of inconvenient recovery of adsorbent.
  2. Please do the experiment on recycling effect of the adsorbents.
  3. More characterization is needed to compare before and after adsorption of Cd2+ by the adsorbent

Author Response

The above four suggestions are very good and enlightening.The question asked is exactly what we hope to solve .

Q1: The mechanism of ion exchange between adsorbent and Cd2+ need to be described in detail.

A1: The mechanism of ion exchange between adsorbent and Cd2+ is difficult, SEM and PXRD analysis after adsorbing Cd2+ by adsorbent found that Cd (CO3)2 may be formed on the surface of the adsorbent. We are not sure about the results of the analysis, so more research is needed. This article focuses on the efficient removal of Cd2+ from aqueous solutions, therefore, we don't think it is necessary to add unconfirmed things to the manuscript.

Q2: How to consider the disadvantage of inconvenient recovery of adsorbent.

A2: The recoverability of dust adsorbent is relatively poor. In the follow-up study, we will consider making a microsphere adsorbent to improve the poor recoverability of adsorbent.

Q3: Please do the experiment on recycling effect of the adsorbents.

A3: The poor recoverability of dust adsorbents is well known. So we don't think it is necessary to do the experiment on recycling effect of the adsorbents.

Q4: More characterization is needed to compare before and after adsorption of Cd2+ by the adsorbent.

A4: We do SEM and PXRD analysis to compare before and after adsorption of Cd2+ by the adsorbent, but it turned out to be unsatisfactory. The purpose of this manuscript is to emphasize the efficient adsorption on Cd2+ by SiO2-Mg(OH)2, so we don't think it is necessary to do more characterization to compare before and after adsorption of Cd2+ by the adsorbent.

Thank you  for your suggestion, which is very enlightening and helpful for us in the future

Reviewer 2 Report

I give the comments corresponding to the line numbers in the publication. The bold marked words and letters have to be included:

line 16: and pH-value on the adsorption.......

line 17/18: ....and pore volume increased by 60,09 %, ........, respectively compared to natural sepiolite.

line 21: .....SiO2-Mg(OH)2 regarding Cd(II)......

line 27: Introduction

line 32   : ....cadmium in wastewater......

line 36: free place between: Cd(II). At.....

line 40: Is it okay to say generally that natural clays show good chemical adsorption?

line 44: "crystal structure of 2: 1 chain clay" What shall this mean?

line 65: [24] loaded....

line 66: ....materials. The effect.....

line 67: [25] studied .........

line 70: ..., biocompatibility......

line 71: ....[26] carried......

line 76: ....were obtained......

line 89:...... Yao et al. [26]. Specific steps .....

line 90: .....performing acid.....

line 101: . Infrared spectrum......

line 103: Batch adsorption experiments

line 108: "and perform shock adsorption of 150 r/min" What does it mean?

line 115: For the kinetic studies......

line 116:   ....and NaOH.....

line 118: After shaking for a certain time (...) at constant temperature the adsorption .....was recorded.

line 119/120: The adsorbed amount is plotted as a function of the adsorption time.

line 121: .....pH-value

line 123/124: ....."a constant temperature oscillation adsorption time" What does it mean?

line 124: When the adsorbent dosage is.......

line 126/127: A proposal  for a different wording for the two sentences: Varying the pH value between 2 and 7 by keeping the other parameters constant, the effect of the pH value on Cd(II) adsorption is evaluated. 

line 134: 3. Results

line 140: .....ammonia is shown.

line 142: increased pore volume? diameter? is missing

line 144: as well as of the ....

line 145: ...material, which are......

line 150: .... are shown.....

line 151: .... there is a large number.....

line 153/154: Is the accuracy of the method for the determination of the specific surface and  as high, that it is possible to mention two decimal places?

line 155: Up to now you always said sepiolite. Suddenly you only use "sep". You have to introduce this abbreviation before.

line 162: The wave numbers range ....

line 164: Is Mg3OH right?

line 177/178: ...((JJCPDS 13-0595) for (110), (130), (060), (131), (260), (080) and (371)) they correspond to sepiolite.

line 196: ...formulae are used as...

line 198: ....... and.....

line 203: ..., which is......

line 204: .... adsorption capacity for ....

line 205: ....larger specific surface area, so the number... . It is not possible to say that because of greater pores more Cd is adsorbed.

line 207/208: "The initial rapid adsorption may be due to the external diffusion of Cd(II) on the outer surface of the material. This process is mainly ion exchange and physical adsorption."The time resolution is low, so that it is not possible to say that the initial adsorption increase is only due to external adsorption. The mass transfer into the porous structure might be fast enough to cause the steep adsorption increase at the beginning.

line 209: .... free active sites.....

line 210: ....in the solution

line 211: ... the resistance....

line 220: Table 1. Kinetic ...

line 229/230: This is because with the increase of the amount of adsorbent added, the relative specific surface area of the adsorbent is increasing, which means that the number of ...

line 231: ...., so that the removal .....

line 232: However, when the amount of the adsorbent exceeds a certain concentration, the increase of the amount of the adsorbent increases the total adsorption sites in the solution rapidly, and some of the adsorption
sites of the adsorbent cannot be completely occupied.

line 235: ..adsorbed by the unit adsorption site is lower, which is shown by a decrease in the unit adsorption. What do you mean with "unit"?

line 236: In the pH range from 2 to 7 .....

line 259: Figure 5. Equilibrium....

line 260/261: than the R2 values of the Freundlich.......

line 269: .....adsorption of single-layer heterogeneous surfaces on heterogeneous adsorption ......incomprehensible

line 274: ... becomes higher. An increase of the adsorption capacity with increasing temperature is typical for chemisorption processes.

line 275: ....Sips model....

line 276/277: .....it can be seen that SiO2-Mg(OH)2 is has compared with similar adsorbents has very good adsorption performance.

line 281/282: Thermodynamic parameters (the standard free energy change â–³G°ï¼Œ standard enthalpy changeâ–³H° and standard entropy change â–³S° )

line 291: The determination of the limit between physisorption and chemisorption is fairly arbitrary: in Bond: Heterogeneous Catalysts: Principles and Applications (1987) a value for this limit is: 

<80 kJ/mol Physisorption

80 kJ/mol < Chemisorption < 800 kJ/mol

A value of 418 kJ/mol should be questioned and a reference should be given, which could be read by english speaking people and which is publicly available.

line 295: ... chemisorption.

line 297: Table 4: unit of the entropy: [kJ/mol K] and for all parameters kJ

line 299: ...nanocomposite modified based on sepiolite

line 300: ...surface area and can be used as...

line 302: The kinetics adsorption isotherm data of the.......

line 305: Therefore, this kind......

Author Response

Response to Reviewer 2 Comments

I have all modified the bold marked words and letters.Special points are as follows:

Points 1: line 40: Is it okay to say generally that natural clays show good chemical adsorption?

Response 1:It has changed to "natural clays expected to become cheap adsorbents for water treatment"

Points 2: line 44: "crystal structure of 2: 1 chain clay" What shall this mean?

Response 2: It has changed to "the crystal structure of 2: 1 type clay minerals"

Points 3: line 108:"and perform shock adsorption of 150 r/min" What does it mean?

Response 3:  It has changed to "and perform oscillating adsorption at a rate of 150 r/min"

Points 4: line 123/124: ....."a constant temperature oscillation adsorption time" What does it mean?

Response 4: It has changed to”a oscillation adsorption time”

Points 5: line 153/154: Is the accuracy of the method for the determination of the specific surface and  as high, that it is possible to mention two decimal places?

Response 5:The results of the commissioned inspection showed that it should be ok.

Points 6: line 235: ..adsorbed by the unit adsorption site is lower, which is shown by a decrease in the unit adsorption. What do you mean with "unit"?

Response 6:"unit adsorption" has changed to "adsorption unit". it shows as a very small adsorption site.

Points 7: line 269: .....adsorption of single-layer heterogeneous surfaces on heterogeneous adsorption ......incomprehensible.

Response 7: "single-layer heterogeneous surfaces"  has changed to "monolayer heterogeneous surfaces"

Additional changes are as follows:

line 307-314 It has added Author Contributions and Conflicts of Interest.

                   "Acknowledgments" has changed to "Funding".

Round 2

Reviewer 1 Report

accept